# Semantics of Dairy Fermented Foods: A Microbiologist’s Perspective

**DOI:** 10.3390/foods11131939

**Published:** 2022-06-29

**Authors:** Francesco Vitali, Paola Zinno, Emily Schifano, Agnese Gori, Ana Costa, Carlotta De Filippo, Barbara Koroušić Seljak, Panče Panov, Chiara Devirgiliis, Duccio Cavalieri

**Affiliations:** 1Institute of Agricultural Biology and Biotechnology (IBBA), National Research Council (CNR), Via Moruzzi 1, 56124 Pisa, Italy; francesco.vitali@crea.gov.it (F.V.); carlotta.defilippo@ibba.cnr.it (C.D.F.); 2Research Centre for Agriculture and Environment, CREA (Consiglio per la Ricerca in Agricoltura e l’Analisi dell’Economia Agraria), Via di Lanciola 12/A, 50125 Florence, Italy; 3Research Centre for Food and Nutrition, CREA (Consiglio per la Ricerca in Agricoltura e l’Analisi dell’Economia Agraria), Via Ardeatina 546, 00178 Rome, Italy; paola.zinno@crea.gov.it (P.Z.); emily.schifano@uniroma1.it (E.S.); 4Department of Biology, University of Florence, Via Madonna del Piano 6, 50019 Sesto Fiorentino, Italy; agnese.gori@unifi.it (A.G.); anamrcosta96@gmail.com (A.C.); 5Computer Systems Department, Jozef Stefan Institute, Jamova Cesta 39, 1000 Ljubljana, Slovenia; barbara.korousic@ijs.si; 6Department of Knowledge Technologies, Jozef Stefan Institute, Jamova Cesta 39, 1000 Ljubljana, Slovenia; pance.panov@ijs.si

**Keywords:** food microbiome, nutrients, food web, metabolic network, nutrition, human health, ontological model

## Abstract

Food ontologies are acquiring a central role in human nutrition, providing a standardized terminology for a proper description of intervention and observational trials. In addition to bioactive molecules, several fermented foods, particularly dairy products, provide the host with live microorganisms, thus carrying potential “genetic/functional” nutrients. To date, a proper ontology to structure and formalize the concepts used to describe fermented foods is lacking. Here we describe a semantic representation of concepts revolving around what consuming fermented foods entails, both from a technological and health point of view, focusing actions on kefir and Parmigiano Reggiano, as representatives of fresh and ripened dairy products. We included concepts related to the connection of specific microbial taxa to the dairy fermentation process, demonstrating the potential of ontologies to formalize the various gene pathways involved in raw ingredient transformation, connect them to resulting metabolites, and finally to their consequences on the fermented product, including technological, health and sensory aspects. Our work marks an improvement in the ambition of creating a harmonized semantic model for integrating different aspects of modern nutritional science. Such a model, besides formalizing a multifaceted knowledge, will be pivotal for a rich annotation of data in public repositories, as a prerequisite to generalized meta-analysis.

## 1. Introduction

Foods yield benefits (or eventually damage) to human physiology that are greater than the benefits of the sums of their ingredients with the nutrients, elements, and other molecules therein included. Most foods are consumed after several modification processes (i.e., cooking or fermentation). During these processes, some molecules and metabolites originally present in the raw ingredients are transformed into bioactive molecules of the final food product by a set of chemical, chemico-physical (i.e., the Maillard reaction during meat cooking), or biological (i.e., fermentation by bacteria or yeasts) reactions. Moreover, the transformations mentioned above also modify the structure of the food matrix, thus improving the bioavailability of some bioactive molecules contained therein (i.e., micronutrients, such as minerals, vitamins, and phytochemicals) [1,2]. This complex set of bioactive molecules and how they change during food preparation ultimately exert potential health effects upon consumption. 

One of the most ancient and widespread food modification processes is represented by fermentation, in which the biological action of microorganisms is responsible for the food modifications [3]. The world of fermented foods is highly heterogeneous, encompassing a broad range of foodstuffs from dairy, meat, fish, vegetable sources, legumes, cereals, and fruits, characterized by distinct production processes and consumption frequencies, reflecting local assets and traditional dietary profiles [4,5]. Fermented foods are significantly represented in the Mediterranean diet and Asian countries. A recent definition of this complex food category, provided by an expert panel of the International Scientific Association for Probiotics and Prebiotics (ISAPP), describes them as “foods made through desired microbial growth and enzymatic conversions of food components” [6]. This definition comprises all foods and beverages obtained through fermentation, irrespective of the presence of living microbes within the food matrix at the time of consumption. 

The multi-faceted body of information, knowledge, and data in the field of nutritional sciences is increasingly being modelled and managed with ontologies and semantic tools [7]. Ontologies are used to specify a shared conceptualization of a domain of interest, and as such, they enable data interoperability, efficient data management and integration, and cross-database search. This trend of using ontologies and semantic tools should not be surprising, given that the interoperability of data, vocabularies, and models in the food sector is of primary interest to consumers and stakeholders at many different levels of the food chain. Among the numerous notable examples of semantic artifacts applied to nutrition, the FOODON [8] and the coordinated initiative which was born among the various ontologies making reuse of the FOODON (https://github.com/FoodOntology/joint-food-ontology-wg, accessed 9 March 2022), inspired by the broader principles layered out by the OBO Foundry [9], are maybe the most interesting for the food domain. Nevertheless, to date, no resources have focused on modeling the technological and biological processes characterizing food fermentation. 

Since its first publication in 2018, the Ontology for Nutritional Studies (ONS) [10] has been committed to taking on the complex task of representing a one-step solution for the annotation and standardization of the multi-faceted nature of nutritional studies. In the present work, we aim at giving our microbiologist’s perspective on what food fermentation implies in a semantic context by recognizing the fundamental work that microorganisms have been doing for us in the production of fermented foods for centuries. We focused our analysis on dairy fermented foods, given their popularity and worldwide diffusion [11]. For this purpose, we chose two use cases, namely kefir and Parmigiano Reggiano [12,13], since they represent a fresh dairy fermented food (kefir) and a ripened cheese (Parmigiano Reggiano), respectively. Indeed, the production steps and the complexity of the involved microbial metabolic pathways are deeply different between these two dairy fermented foods. On one hand, the fresh product relies on sugar fermentation performed in the early stages by microbial starter cultures. The ripened product, on the contrary, is also characterized by additional transformations directed at proteins and lipids, which are responsible for the ripening process and are usually performed by non-starter microbes. 

It is sometimes common for modern nutritional studies to include metagenomic data (i.e., presence of specific genes in the microbial community, microbiome), meta-taxonomic data (i.e., characterization of the composition of the microbial community, microbiota), and metabolomic data (i.e., characterization of bodily fluid metabolites, formed following the consumption of certain foods). All those facets are intimately connected and influenced by the consumption of fermented foods, hence considering them as independent entities is an oversimplification of our ability to understand the complex interactions between health and diet. In this work, we specifically aimed at providing solutions to overcome the problems connected to such an oversimplification. 

Specifically, our aim was to provide an ontological resource that (1) builds a clear conceptual framework for the formalization and sharing of information in the dairy fermented food domain; by determining the terminology (i.e., concepts and their definitions) and how those concepts are connected among each other, and (2) set the basis for the integration of data in nutritional studies, increasingly characterized by the use of multi-omics techniques; by connecting in the same model the compounds in starting ingredients to the metabolites in the final product (relating to metaproteomics and metabolomics), with the genes and pathways directing those transformations (related to metatranscriptomics), and finally with the organisms which carry those genes and performs those transformations (related to metagenomic). 

## 2. Materials and Methods

For the extension of Ontology for Nutritional Studies (ONS) with an ontological representation of the fermented food model, we followed the same logic and methods used for the initial development of ONS [10]. A vocabulary of the domain (i.e., a set of terms required for knowledge representation of the fermented food domain) was initially constructed by multiple interactions and discussions among the authors of this manuscript. The vocabulary was then organized in a hierarchical structure of classes and subclasses, and finally contextualized with the knowledge model of ONS. 

As done previously, we adhered to the OBO Foundry Principles [9] during development. In compliance with the orthogonality principle, we checked for the presence of each of the terms in our vocabulary (or for the presence of a suitable synonym) among other ontologies in the OBO Foundry repository. ONTOFOX [14] was used to import those terms and their annotations from original ontologies. In addition, newly defined concepts were defined and included as previously reported, with a label formed by an “ONS_” prefix followed by a 7-digit number. Table 1 summarizes the source ontologies (and their corresponding bibliographic references) of terms imported into ONS as a result of this procedure and to enable the modeling presented in this manuscript. Besides the ontology name, we list the prefix (referenced in the text and figures), the web reference of the resource, and, if available, the citation to the paper that describes the ontology.

Accessibility and traceability of the development of the presented resource were ensured by versioning and development using the GitHub functionality and licensed under the CC-BY 4.0 license. Master ONS repository can be found at https://github.com/enpadasi/Ontology-for-Nutritional-Studies, but the ontology is indexed in all principal ontology browsers (i.e., ONTOBEE http://www.ontobee.org/ontology/ONS, OLS https://www.ebi.ac.uk/ols/ontologies/ons, BIOPORTAL https://bioportal.bioontology.org/ontologies/ONS, all link accessed on 28 April 2022) and is part of the OBO foundry library (https://obofoundry.org/ontology/ons.html, accessed on 28 April 2022).

## 3. Results

### 3.1. Fundamental Concepts of Fermented Foods

Firstly, the class pertaining to the high-level concept of food fermentation was imported from FOODON ontology [8] [FOODON:00001304; def. “A fermentation process in which either carbohydrates, proteins or fats are modified through microbial, enzymatic and/or other biological processes’’]. From a microbiological perspective, we detailed this conceptually high-level process in different sub-processes generally applicable to any (dairy) fermented food (Figure 1). This first level of classification deals with how the microbial metabolism transforms a variety of molecules present in the raw ingredients to finally generate the taste, aroma, texture, and safety profile characterizing the fermented final product (Figure 1). 

From a semantic point of view, this connection between the whole process (i.e., food fermentation) and its parts was defined with the property “has component process” [RO_0002018]. In contrast, the activity of starter and non-starter cultures, and their connection with different sub-processes of the fermentation, was defined with the property “participates in” [RO_0000056]. Hence:“starter culture for food fermentation” participates in “glucose metabolism during food fermentation”“food fermentation” has component process “glucose metabolism during food fermentation”

The three main metabolic pathways involved in metabolite production in the dairy fermentations are: the fermentation of sugars, starting from glycolysis and leading to the production of lactic acid, is generally performed by lactic acid bacteria (LAB) [ONS:0000142; “glucose metabolism during food fermentation”, the set of reactions leading to the conversion of glucose into pyruvic acid, which is a substrate for the fermentation processes]. Through the glycolysis pathway, obligate homo-fermentative LAB can convert glucose into pyruvic acid, which is then converted to lactic acid, the main product of the fermentation process [22]. On the other hand, hetero-lactic fermentation is characterized by the formation of different co-products such as CO_2_, ethanol, and/or acetic acid in addition to lactic acid. As described above, lactic acid is the main product, but a fraction of pyruvic acid, the glycolysis end-product, can alternatively be converted to diacetyl, acetoin, acetaldehyde, or acetic acid, contributing to flavors characterizing several dairy products, such as yogurt and butter;the lipolysis [ONS:0000140; “lipolysis during food fermentation”, the set of reactions leading to the conversion of fats (triglycerides) into free fatty acids, monoacylglycerol, diacylglycerol, and glycerol during the fermentation process]. Regarding fat metabolism, the degradation of milk fats releases free fatty acids and glycerol, monoacylglycerols, or diacylglycerols. Fatty acids are essential to determine flavor in foods such as cheeses, and they can also act as precursors of other flavor compounds, including lactones [23];the proteolysis [ONS:0000141; “proteolysis during food fermentation”, the set of reactions leading to the hydrolysis of milk proteins into peptides during the food fermentation process]. Proteolysis is the main process influencing the rate of flavor and texture of fermented foods, and it usually occurs during the maturation stage of food production. Typically, LAB growth requires the hydrolysis of milk proteins since they cannot synthesize many amino acids, vitamins, and nucleic acid bases [24]. The hydrolysis of milk proteins into peptides is due to LAB proteolytic enzyme activity. Exopeptidases and endopeptidases further hydrolyse the resulting peptides to small peptides and amino acids [25].

Despite a distinction based on the type of molecule being metabolized, those classes are also characterized by differences in technological aspects pertaining to the moment they happen relative to the whole fermentation and maturation process, defined by the microbial community involved. Glucose metabolism is mainly performed in the initial phases of fermentation by the starter cultures [ONS_0000134], while protein and lipid metabolism are typically involved in the maturation of fermented foods [ONS:0000165, “cheese maturation”, def. the process in which the shaped cheese is stored in controlled temperature conditions and for a defined amount of time to give rise to the ripened cheese] and are among the main pathways responsible for the development of secondary metabolites connected to sensory characteristics of the final product. 

Those pathways are typically carried out by non-starter microbial cultures [ONS:0000135]. The world of fermented foods is very diversified, and differences can also be highlighted in the type of starter culture employed. First of all, fermentation can be performed by the microbiota that is usually part of the raw matrix, i.e., microorganisms naturally present in the raw food or processing environment [ONS:0000124, “autochthonous food microbiota”]. On the other hand, pasteurization or other treatments kill microorganisms, and fermentation must be started by the intentional addition of starter cultures. These can be “commercial starter cultures” [ONS_0000123], namely microorganisms selected for specific technological and sensory properties that are inoculated directly into food materials used to initiate the fermentation process and in order to bring about desired and predictable changes in the finished product [26]. Starter cultures are commercially available in liquid, frozen, or lyophilized form from several companies serving global markets.

On the contrary, “natural starter cultures” [ONS:0000122] represent unselected, autochthonous microbial consortia that initiate the fermentation, often employed in the “backslopping” process [ONS:0000127], which consists of the addition of a small amount of a previously fermented batch to the raw food, for example during sourdough bread preparation [27]. We further distinguished between starter culture and “microbial inoculum” [ONS:0000125] role, defined as “the role of a material entity that is the vehicle for the introduction of one or more microorganisms (representing starter culture) into a suitable substrate to initiate the fermentation process”. The fermentation process itself is always performed by a ‘starter culture’, but the vehicle of such a community (i.e., ‘microbial inoculum’) can be different. For example, in kefir fermentation, this modeling was useful to account for the fact that fermentation is started by the addition of ‘kefir grains’ [ONS:0000126] in milk through a ‘backslopping process’. Kefir grains represent a complex matrix composed of a polysaccharide associated with bacteria and yeasts. From a semantic point of view, kefir grains are not independent continuant but rather generically dependent continuant, as they are formed (i.e., depend on) due to kefir fermentation, arising a chicken-or-egg-like problem. For further details on the kefir fermentation, see the dedicated section of the manuscript.

### 3.2. Semantic Representation of Microbial Metabolism in Food Fermentations

The classification already introduced and depicted in Figure 1 was further detailed, expanding on the specific microbial metabolic pathways involved in the above-mentioned sub-processes of the food fermentation flow. This is indeed a vast and very diverse topic, as a multitude of different molecules deriving from microbial primary and secondary metabolism are produced within different fermented foods. Consequently, different metabolic pathways are fundamental for the production of metabolites responsible for the final taste, texture, aroma, and health features of fermented foods. In this manuscript, we presented the first semantic formalization of such complex knowledge, scratching the surface of the knowledge domain of food fermentation, by focusing on the key metabolism involved in dairy products: glucose metabolism. 

We must duly premise that food fermentation is carried out by a complex community of microorganisms. Moreover, most substances are co-metabolized by distinct microorganisms, and it is challenging to model temporal and consequential relationships between concepts specifically. Thus, at the moment, we focused our activities on modeling fluxes between ‘input’ and ‘output’ of the different metabolic pathways, combining whenever possible GO pathways with CHEBI chemical entities, and on the sequential relationships between different GO pathways.

Part-to-whole modeling. At the first level, we divided the whole process into mandatory sub-processes. Then, as already done for the main classification of fermentative metabolisms, we used the “has component process” relation [RO_0002018] to connect the fermentative pathway or process [i.e., “homolactic fermentation” GO_0019661] to the set of modeled sub-process. To do so, we followed the following principles: (I) including preferably those pathways already available and defined in GO, and (II) including all pathways capable of explaining the metabolism from the compound found in the raw material being fermented to the final products.Input/output modeling: at the second level, we modeled which compound (i.e., CHEBI class) was to be considered as input (i.e., the starting material entity on which acts a specific GO pathway) or output (i.e., the material entity resulting from the action of a GO pathway) of each of the outlined steps of the model.Consequential modeling. We indicated the consequential relation between pathways using the “causally upstream of” property in RO [RO_0002411]. It is very common for biological pathways to be interconnected, with one pathway acting on the end products of the other, which are connected in a causal chain.

An example that recapitulates all the above-mentioned modeling can be found in the “homolactic fermentation” concept [GO_0019661, label: glucose catabolic process to lactate via pyruvate]. This class has the more general label of “glucose catabolic process to lactate via pyruvate” in the GO, and the “homolactic fermentation” is annotated as a synonym. It is defined as “The anaerobic enzymatic chemical reactions and pathways resulting in the breakdown of glucose to lactate, via canonical glycolysis, yielding energy in the form of adenosine triphosphate (ATP)”. 

From this definition in GO, we were able to identify the suitable input and output compounds of the pathway, which would be glucose and lactic acid, respectively. Those were connected, as discussed above, with the “has input” and “has output” relations. The “glucose catabolic process to lactate via pyruvate” pathway can be viewed as a general pathway composed of at least two interconnected and fundamental sub-processes in dairy fermentations. On one side, there is firstly the biological need to release glucose from lactose before it can be the input of some other metabolic pathway, thanks to microbial lactose hydrolysis activity (beta-galactosidase activity; [GO:0004565]). Then, the process of “canonical glycolysis” [GO:0061621] can be connected as an integral part of the more general concept of the catabolic process. As defined in GO, we collected those classes and connected them with part-to-whole, input/output, and consequential modeling as described above, as represented in Figure 2.

Briefly, the general “homolactic fermentation” [GO:0019661] “has component process” at least in two sub-processes. The general input of this pathway is glucose [CHEBI:17234], while the general output is lactic acid [NCIT:C76926]. The first mandatory sub-process involves “beta-galactosidase activity” [GO:0004565], leading to the breakdown of lactose [NCIT:C28166] into glucose [CHEBI:17234] and galactose [CHEBI:28260]. The glucose can then be the input of the canonical glycolysis pathway [GO:0061621], finally resulting in pyruvate [NCIT:C116012]. Those two pathways are consequential, with the lactose hydrolysis necessary to release glucose, and generally preceding the glycolysis [GO:0004565 causally upstream of GO:0061621]. 

A more comprehensive view of the diverse glucose metabolic pathways involved in dairy food fermentation is presented in Appendix A. Besides lactic fermentation by bacteria, glucose can be fermented to ethanol and carbon dioxide through alcoholic fermentation by the yeast’s metabolism, with implications for food preservation and taste. While being fundamental for the final dairy fermented food taste, texture, preservation, and consumption, they will not be covered in this manuscript as all the models we developed are centered on the glucose metabolism and the lactic fermentation by bacteria as application scenarios.

Presenting a detailed view of the complexity of the process involved in food fermentation is beyond the scope of this manuscript. Here rather, thanks to the use of two use cases, we have focused our attention on the modeling of some particular aspects: (I) the initiation of a dairy fermentation process and the role of inoculum; (II) the prevalent composition of fermented food microbiota and mycobiota.

#### 3.2.1. Kefir Use Case: Fresh Dairy Fermented Food

Kefir, which originated in the Caucasus Mountains, is traditional fermented milk containing over 50 species of microorganisms, including LAB, yeasts, and acetic bacteria, identified from a spontaneous microbial starter culture known as kefir grains [28]. 

For kefir, a new fermentation process can be initiated by adding a “starter culture for food fermentation” [ONS:0000134; microbial culture responsible for the beginning of the fermentation process], either natural or industrially selected and produced, to the raw ingredient, namely the milk. This starter culture constitutes the inoculum for fermentation [ONS:0000146; vital biomass containing microorganisms added to start the fermentation process], which physically brings microorganisms to the raw milk ingredient. Kefir can be obtained starting from cow, sheep, goat, or buffalo milk. As a result of this first fermentation batch, kefir food products [NCIT:C173982] and kefir grains [ONS:0000126; complex matrix composed of an exopolysaccharide in association with bacteria and yeast cells in kefir] are produced. The latter, once recovered, can be used as a microbial inoculum for subsequent fermentation processes [kefir grain as inoculum, ONS:0000147, complex matrix composed of an exopolysaccharide in association with bacteria and yeast cells added to milk for kefir production], through a “backslopping” procedure [ONS:0000148, the procedure in which a small amount of the finished or intermediate fermentation product is used to inoculate a new batch to start a new fermentation process]. This modeling highlights the cyclic nature of dairy fermentative processes, where the backslopping process is widely used for the continuation of the iterative fermentative process. The model of kefir fermentation is described in Figure 3.

The kefir microbiota [ONS:0000144, consortium of living microorganisms present in kefir, characterized by bacteria and yeast components (has part)] can be found both in the final product [NCIT:C173982, “kefir”] and in the kefir grains [ONS_0000126], and display a similar composition (see boxes in Figure 3, reporting the prevalent genera that we retrieved from the literature review in each entity). In our modeling, specific instances of the various genera classes/concepts that can be found in the NCBITaxonomy ontology were linked to the concept of kefir microbiota (which is an “inoculum for fermentation” [ONS:0000146] and participates in the “glucose metabolism during food fermentation” [ONS:0000142; the set of reactions leading to the conversion of glucose into pyruvic acid, which is a substrate for the fermentation processes]) using the “has active ingredient” property (Figure 3).

#### 3.2.2. Parmigiano Reggiano Cheese Use Case: Ripened Dairy Fermented Food

The technological aspects connected to the Parmigiano Reggiano (PR, a hard, cooked, and slow-maturing cheese) production are considerably more complex than those reported for kefir fermentation. Its inclusion in the list of cheeses bearing the protected designation of origin (PDO, EU regulation 510/2006) poses restrictions to its geographic area of production and its technological characteristics. In particular, a considerable difference stands in the initiation of the whole fermentation process. As already introduced, like for other dairy fermented food products, Parmigiano Reggiano fermentation (Figure 4) is initiated by the addition of the raw cow’s milk, partially skimmed through natural cream surfacing (“starting milk mixture” [ONS:0000170; def. the mixture obtained by combining milk from morning milking with milk from previous evening milking and used as an initial ingredient in Parmigiano Reggiano making]), of a certain product or sub-product of the previous fermentation batch. In addition, the PDO designation dictates that milk (“starting milk mixture” [ONS:0000170]) may not be subjected to heat treatments and must derive from cows whose diet is based on the use of fodder obtained in the area of origin. 

The whole process is thus iterative in time. Notwithstanding, a detailed set of rules in the PDO prevents the initiation of a new fermentative process. In kefir fermentation, a new fermentation process can be initiated at any moment through the use of an inoculum, even in the absence of a backslopping from the previous batch. Conversely, in Parmigiano Reggiano, no allowed inoculum can be employed to start a new fermentation process. Instead, each batch of fermentation must be initiated uniquely by the addition of whey (“whey as inoculum” [ONS:0000175]) obtained by the previous fermentation, in the backslopping process [ONS:0000148]. Indeed, natural whey starters (“whey as inoculum” [ONS:0000175], the so-called “siero innesto naturale”), which initiate fermentation, are obtained from the spontaneous acidification of the whey remaining from the manufacture of the previous day, used as inoculum for backslopping. The microbiota of whey starter (“whey microbiota” [ONS:0000150]; consortium of living microorganisms present in whey, characterized by bacteria and yeast components (has part)) is characterized by multiple thermophilic LAB strains, which have a higher chance of survival when subjected to the selective pressures represented by the heating procedures. Moreover, PR is a slow-maturing cheese, which implies that the final food product (“parmigiano reggiano” [ONS:0000163]) exists exclusively as the output of a “parmigiano reggiano maturation” process [ONS:0000166] lasting at least 12 months. The final product, aged 12, 18, 24, 30, or over 40 months, can be marketed in whole forms, in portions, or grated. 

Regarding microbiota composition, the technological parameters such as the cooking temperature of the curd, the grafts’ temperature, and cooling mode, and different acidity and pH values can influence the bacterial consortium [29]. During cheese ripening, milk caseins are hydrolyzed by the action of different proteases, deriving from milk and rennet, and of microbial origin, [30]. The proteolysis produces oligopeptides and free amino acids. During the maturation, also lipolysis takes place due to the activity of lipolytic enzymes esterases, and lipases, which convert triglycerides into free fatty acids (FFA), monoacylglycerol, diacylglycerol (DAG), and glycerol. In particular, FFA are vital components in PR, since they contribute directly to its flavor and act as substrates in different reactions producing alcohols, aldehydes, and lactones, among other molecules that influence cheese flavor [31].

### 3.3. Modeling Nutritional, Technological, and Safety Aspects Impacting Human Consumption

As stated in the Introduction, foods exert benefits (or eventually damage) to human health that are greater than the benefits of the sums of their ingredients. This is especially true in the diverse world of dairy fermented food products, in which the fermentation technological process has threefold implications:As a consequence of microbial metabolism, the chemico-physical properties of the raw ingredients are modified. For example, during dairy food fermentation, the lactose contained in the starting raw ingredient (milk) is gradually metabolized, leading to the accumulation of lactic acid (see Figure 2).As a consequence of microbial metabolism, the chemical composition of the raw ingredients is modified by lowering the concentration of certain chemical compounds and increasing the concentration of final (microbial) metabolites.The microbiota composition of the final fermented food product is the result of selective pressure imposed by both technological and microbiological factors.

An example of the changes in chemico-physical parameters that happen during dairy food fermentation, and their implications, is reported in Figure 5A. In this modeling, reused from the modeling in the OBI ontology [32], the concentration of lactic acid in the final product can be measured, producing a particular measurement datum [IAO:0000109] with a pH-based unit [UO:1000196] of a specific value (4.5 for the final kefir product). In this context, this specific pH value has the role of “agent for physico-chemical selection of microbiota” [ONS:0000159, def. “a chemical and/or physical factor that exerts selection pressure and brings about natural selection within microbiota components”], which is determined by the ability of different bacterial strains to grow at acid pH values (i.e., “agent for physico-chemical selection of microbiota” determined by “altered effects of pH on population growth” [OMP:0007138]). Furthermore, the selection of an acidophilic microbiota has implications for the preservation of fermented food (“agent for physico-chemical selection of microbiota” inheres in “food preservation process” [FOODON_03470107]), as microorganisms connected to contamination and/or spoilage of food are generally unable to grow at those pH values. 

Therefore, instances of the data item for the dairy fermented food reality modeled in this manuscript, were included under the “measurement datum” class (IAO:0000109) using class numbering starting with number 9. In detail, we created “lactic acid concentration in kefir” [ONS:9000001], “lactose concentration in starting milk mixture” [ONS:9000002], “lactose concentration in parmigiano reggiano” [ONS:9000003], “temperature of parmigiano reggiano curd cooking” [ONS:9000004], “duration of parmigiano reggiano curd cooking” [ONS:9000005], “water activity of parmigiano reggiano” [ONS:9000006], and “duration of parmigiano reggiano maturation [ONS:9000007]. As an example, Figure 6 reports assertion and axioms for the individual ONS:9000001 (“lactic acid concentration in kefir”).

With similar modeling, we can represent how the changes in chemical compound/metabolite concentration that happen during dairy food fermentation have implications on consumer health (Figure 5B). The decrease in pH value of the final product is based on the utilization of the lactose contained in milk. For this reason, correspondingly with the decrease of pH due to the accumulation of lactic acid, there is also a decrease in initial lactose concentration. We can imagine assessing the concentration (g/100 mL) of lactose in two entities involved in the Parmigiano Reggiano fermentation: the “starting milk mixture” and the final “parmigiano reggiano”. We would result in this way with two distinct measurement data [IAO_0000109], both with a mass measurement unit [UO:0000021] and a specific value (4.8 for starting milk mixture, <0.01 for the final parmigiano reggiano). The model is similar to what was reported previously, but in this case, those specific values assume the role of being “unfit for consumption” [ONS:0000173, def. “the role of a material entity, generally a chemical compound, such as the material entity is unfit for consumption by a defined group of individuals. 

This role has a causal relation (is determined by) with another material entity such that it exerts a strong causal influence on the role, and its removal causes the termination of the existence of the role”] or “suitable for consumption” [ONS:0000172, def. “the role of a material entity, generally a chemical compound, such as the material entity is not suitable for consumption by a defined group of individuals. This role has a causal relation (is determined by) with another material entity such that it exerts a strong causal influence on the role, and its removal causes the termination of the existence of the role”], respectively. In both cases, those specific roles are determined by the “lactose intolerance” [HP:0004789] of the consumer (determined by [RO_0002507; s determined by f if and only if s is a type of system, and f is a material entity that is part of s, such that f exerts a strong causal influence on the functioning of s, and the removal of f would cause the collapse of s]. If the “lactose intolerance” [HP:0004789] constraint is missing, no evaluation of unfit or suitable for consumption could be performed. 

In Figure 7 and Figure 5A, we show examples of the technological aspects during dairy food fermentation that pose selective pressure and define the final product microbiota. Similar to how the acid pH has a role in the selection of the kefir microbiota and the preservation (i.e., shelf life), the “parmigiano reggiano curd cooking” [ONS:0000164] is a process that has quality-defined values of temperature and time (55 °C and 10 min, respectively, [33]). Those values have the role of “agent for physico-chemical selection of microbiota” [ONS:0000159], which is determined by the different ability of different bacterial strains to grow at high temperature (i.e., “agent for physico-chemical selection of microbiota” determined by “altered population growth at high temperature” [OMP:0007952]) leading to the selection of a thermophilic microbiota (“agent for physico-chemical selection of microbiota” inheres in “parmigiano reggiano microbiota” [ONS:0000167]. Similarly, the addition of salt during the process of “parmigiano reggiano shaping and salting” [ONS:0000168] by a lowering in water activity selects the “parmigiano reggiano microbiota” [ONS:0000167] owing to the “altered population growth in high osmolarity” [OMP:0007519]. 

This modeling again highlights how the technological steps in the two use cases are remarkably different. In fresh dairy fermented food, the community in the final fermented food product is essentially determined by the selection imposed by an acidic pH. In ripened dairy fermented food, on the contrary, the community in the final fermented food product is determined by multiple processes (i.e., cooking, salting, maturing) and imposed by different parameters such as temperature/time combination (during cooking) and water activity (during salting and ripening).

## 4. Discussion

Human nutrition is mediated by the processes that render food ingredients and bioactive food compounds available and transformable. Among those processes, fermentation has played a fundamental role in human evolution by, for example, stabilizing food products and allowing the storage of sources of nutrients that would have been otherwise lost [34]. Moreover, the live microorganisms entering the host’s body through the consumption of fermented foods have provided the ancient human societies, expanding from the rural lifestyle, with the full enzymatic potential required to support their growth. We can undoubtedly state that most of human biochemistry and physiology are tuned to life conditions that existed before the advent of agriculture some 10,000 years ago; genetically, our bodies are the same as they were at the end of the Paleolithic era. Therefore, thanks to the contribution of a far more plastic genetic pool, namely the one of commensal microbes, we can utilize and transform fiber, milk, grains, gluten, and a number of by-products of microbial fermentation evolutionally acting as a holobiont [35,36]. A growing body of evidence demonstrates that fermented foods provide an unprecedented opportunity to extend the number and composition of our commensals, and how diet globalization and the loss of microorganisms from traditional foods are potentially endangering the supply of these microbial communities to the human body [37,38]. The number of fermented food-associated microorganisms that are now part of the standard probiotic supplementation further demonstrates how this research field is expanding [39]. 

These considerations indicate how an appropriate description of foods with correctly defined terms and rich metadata annotation is important from several points of view. Moreover, a semantic resource includes a formalization of the knowledge of the biological and technological processes involved in food fermentation, thus bridging information from very different fields of chemical reactions determining the transformation of raw ingredients to food bioactives, enzymes involved in those reactions, and genes coding for those enzymes, as well as microbial genomes containing those genes, would further increase our ability to represent the real world. From the nutritional and computational standpoint, ontologies are the best and most appropriate instrument for this purpose, as they can categorize and make this information searchable in a coherent way [40]. Indeed, food ontologies are acquiring a central role in human nutrition, allowing us to formally describe the multi-faceted nature of the nutritional domain [41]. A precise determination of the terms describing foods and food production processes is thus more and more central to the internet of things and its integration into nutrigenomics approaches [42,43].

In the present work, we aimed at developing a semantic model for a more comprehensive view of fermented foods. We included concepts ranging from technological elements in the production phases to metabolomic and genomic elements upon their consumption. Emphasis was given to the connection of specific microbial taxa to the various processes and sub-processes of dairy food fermentation. As a result, we were ultimately able to focus on: (I) elucidating the various gene pathways involved in raw ingredient transformation, mainly connecting stand-alone entities in GO to each other, with respective input and output material (substance/molecule from CHEBI or NCIT) and connecting them in a proper sequential order (use of causally upstream relation in the relation ontology (RO)); (II) connecting the final products of microbial metabolic pathways to their consequences on the fermentation process, including improvement in the shelf life (e.g., thanks to acidification) and specific health claims (e.g., low lactose concentration that makes the food suitable for lactose intolerant subjects). Specific taxa were also connected to the microbiota commonly found in the final fermented product, or during different stages of the production chain, resulting from a literature review (using active ingredient connections). 

In particular, we focused on the primary metabolism of carbohydrates, lipids, and proteins to build the basic semantic framework, which may in the future be easily extended to other foods and other microbial metabolic pathways. However, various pitfalls are currently limiting our ability to more in-depth detail these concepts. For example, metabolic pathways are static classes, and we can only model their inputs and outputs or sequentially connect one metabolism to another. This only partially accounts for the real process, where the various metabolic pathways are simultaneously activated by a complex and diverse communities, in a food matrix with changing chemico-physical conditions. In this context, it is not assured that a metabolic pathway will effectively be completed from input to output, given that the intermediates could be redirected to other metabolic pathways, depending on the context of gene regulation.

Future activities will include other foodborne microorganisms and their specific functions in this growing model, with the modeling of other fermented foods, finally aiming at representing the complex ecological and metabolic interplay that occurs in their microbial community. In the case of fermented foods, microorganisms would be the main players in the food preparation process in terms of biological fermentation and could also reach alive and vital the human host gut through consumption. Once completed, this would represent the nexus among molecules in the raw ingredients and their transformation into bioactive metabolites in the food through the action of microorganisms, as well as a complete catalogue of microbial species and molecular functions that the human host takes up following consumption of fermented foods. Using precisely defined ontological terms will facilitate cross-reference, cross-link and will uncover information on fermented foods present on the web or in the public domain to further support the health claims associated with these products.

## 5. Conclusions

Food ontologies are becoming central in human nutrition, allowing us to adequately describe the intervention and observational trials. In the present work, accounting for our background in microbiology, we introduced a further development of ONS revolving around technological, genomic, microbiological, and metabolomic concepts connected to fermented food consumption in order to provide an ontological resource that includes, in addition to conceptualization, the basis for the integration of data in nutritional studies, increasingly characterized by the use of multi-omics techniques. In this way, we demonstrated the potential of ontologies to formalize the various gene pathways involved in raw ingredient transformation, connect them to resulting metabolites, and finally to their consequences on the fermented product, including technological, health, and sensory aspects. Furthermore, our work will be instrumental to the FAIR (Findable, Accessible, Interoperable, Reusable) annotation of the experimental data in public repositories [44]. This process will improve the findability, comparability, and integration of heterogeneous data and the exploitation of publicly available datasets through meta-analyses.

Finally, the developed ontological resource presents a vital cornerstone of the food- and nutrition-related semantics that has been developed by several EU-funded initiatives and projects, such as JPI HDHL ENPADASI, FP6 ISO-FOOD, H2020 Food Nutrition Security Cloud (FNS-Cloud), ELIXIR Food & Nutrition and others. In FNS-Cloud, several services and tools are being developed to demonstrate the applicability of the semantics in the interoperability of food- and nutrition-related data, which is crucial in dealing with open research questions and gaps in evidence on how to transform food systems so that they benefit human nutrition and health, as well as environment. Linking various semantic resources, such as FOODON [8], ONS [10], ISO-FOOD [41], etc., through advanced tools like FoodViz [45], enables deep exploration of data in a transparent and trustworthy way.

## Figures and Tables

**Figure 1 foods-11-01939-f001:**
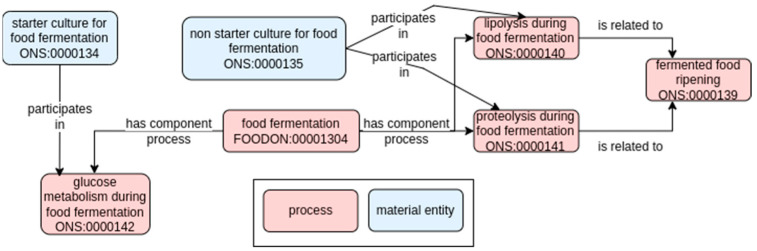
Top-level model for the metabolic pathways involved in dairy fermentation. Solid arrows represent semantic relations in RO (annotated with their label), and pink and blue rectangles refer to process or material entity, respectively. The label and the term ID (formatted as an ontology prefix followed by a number) are reported for each class.

**Figure 2 foods-11-01939-f002:**
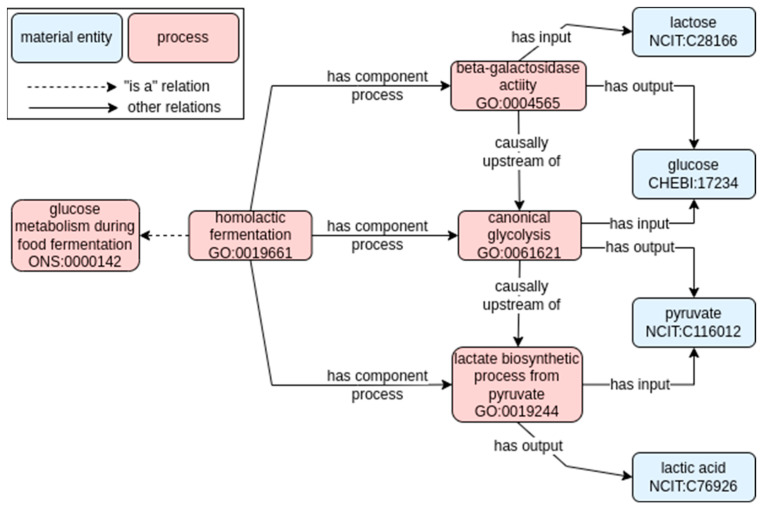
Ontological model of glucose metabolism involved in dairy fermentation. Solid arrows represent semantic relations in RO (annotated with their label), while dashed arrows represent hierarchical relationships (i.e., “is a” relation). Pink and blue rectangles refer to process or material entity, respectively. The label and the term ID (formatted as an ontology prefix followed by a number) are reported for each class.

**Figure 3 foods-11-01939-f003:**
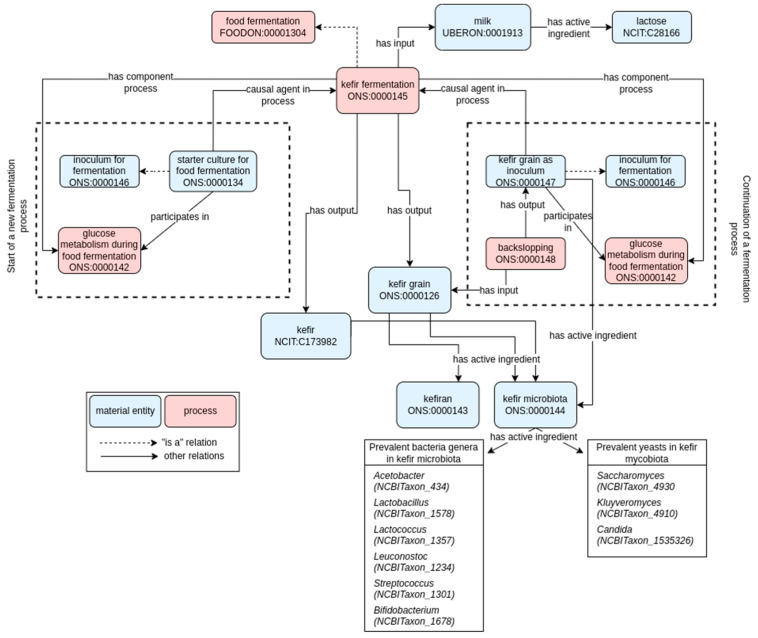
Ontological model of kefir fermentation. Solid arrows represent semantic relations in RO (annotated with their label), while dashed arrows represent hierarchical relationships (i.e., “is a” relation). Pink and blue rectangles refer to process or material entity, respectively. The label and the term ID (formatted as an ontology prefix followed by a number) are reported for each class. Classes are generally conceptualized to the universal level (i.e., the definition and concepts apply to any possible class instance). When suitable, instance-level concepts (indicated with white rhombus) are used to indicate a particular instance of the universal concept. For example, the prevalent bacteria genera in kefir microbiota are specific instances of the universal “bacteria” (or taxonomical subclassifications) concept (i.e., only some specific instances of the Acetobacter genus are active ingredients of the kefir microbiota, but not the entirety of the Acetobacter concept).

**Figure 4 foods-11-01939-f004:**
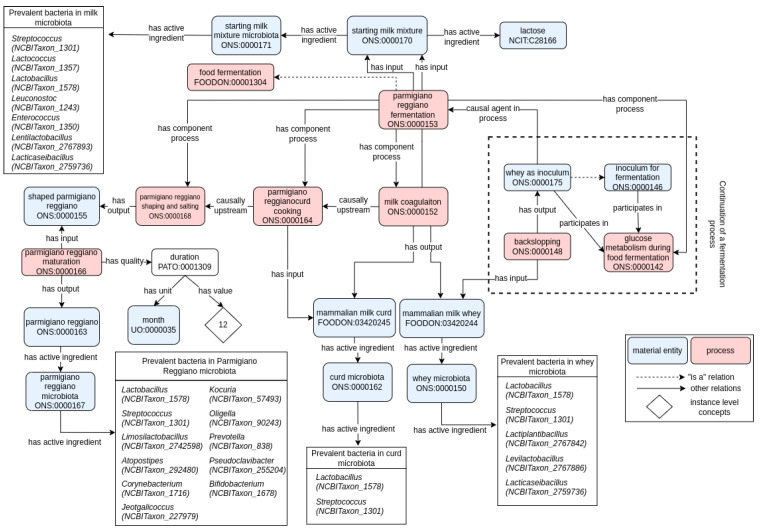
Ontological model of Parmigiano Reggiano fermentation. Solid arrows represent semantic relations in RO (annotated with their label), while dashed arrows represent hierarchical relationships (i.e., “is a” relation). Pink and blue rectangles refer to process or material entity, respectively. The label and the term ID (formatted as an ontology prefix followed by a number) are reported for each class. Classes are generally conceptualized to the universal level (i.e., the definition and concepts apply to any possible class instance). When suitable, instance-level concepts (indicated with white rhombus) are used to indicate a particular instance of the universal concept. For example, the prevalent bacteria genera in Parmigiano Reggiano microbiota are specific instances of the universal “bacteria” (or suitable taxonomical subclassifications) concept (i.e., only some specific instances of the *Lactobacillus* genus are the active ingredient of the Parmigiano Reggiano microbiota, but not the entirety of the *Lactobacillus* concept).

**Figure 5 foods-11-01939-f005:**
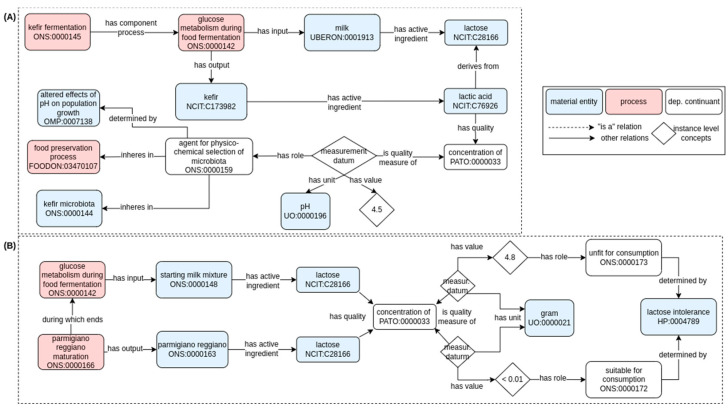
Ontological model depicting some technological (i.e., food preservation) (**A**) and nutritional (i.e., being suitable for consumption to lactose-intolerant subjects) (**B**) services provided by the microbiota in dairy fermented foods. Solid arrows represent semantic relations in RO (annotated with their label), while dashed arrows represent hierarchical relationships (i.e., “is a” relation). Pink and blue rectangles refer to process or material entity, respectively. The label and the term ID (formatted as an ontology prefix followed by a number) are reported for each class. Classes are generally conceptualized to the universal level (i.e., the definition and concepts apply to any possible class instance). When suitable, instance-level concepts (indicated with white rhombus) indicate a particular instance of the universal concept.

**Figure 6 foods-11-01939-f006:**
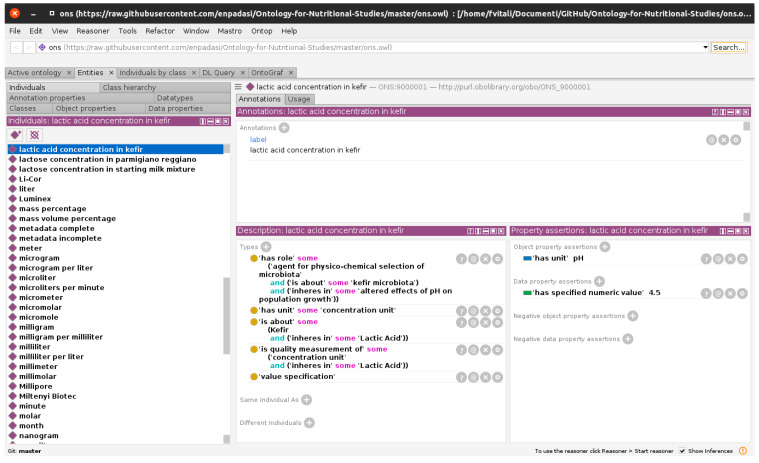
Screenshot taken from the Protégé ontology editing program, showing the assertion and axioms for individual ONS:9000001 (“lactic acid concentration in kefir”).

**Figure 7 foods-11-01939-f007:**
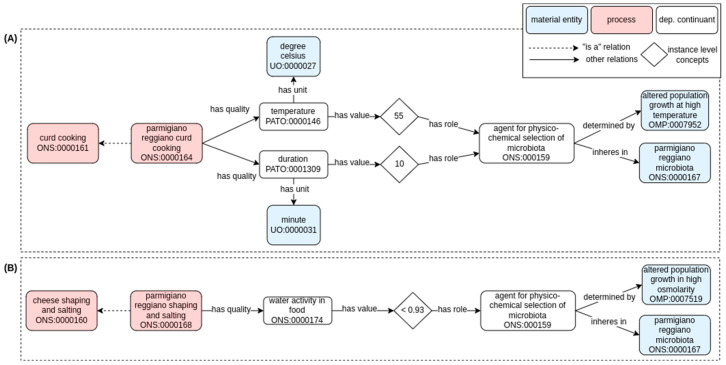
Ontological model depicting some technological steps in dairy food fermentation by imposing selective pressure on the microbial community based on temperature (**A**) or water activity (**B**); fundamental for determining the microbiota composition in the final food. Solid arrows represent semantic relations in RO (annotated with their label), while dashed arrows represent hierarchical relationships (i.e., “is a” relation). Pink and blue rectangles refer to process or material entity, respectively. The label and the term ID (formatted as an ontology prefix followed by a number) are reported for each class. Classes are generally conceptualized to the universal level (i.e., the definition and concepts apply to any possible class instance). When suitable, instance-level concepts (indicated with white rhombus) are used to indicate a particular instance of the universal concept.

**Table 1 foods-11-01939-t001:** Reuse of terms from pre-existing external ontologies, imported in ONS for completing the modeling in this manuscript.

Ontology	Prefix	Web Reference	Citation
Food Ontology	FOODON	https://foodon.org/ (accessed on 28 April 2022)	[8]
Relation Ontology	RO	https://oborel.github.io/ (accessed on 28 April 2022	
Gene Ontology	GO	http://geneontology.org/ (accessed on 28 April 2022)	[15,16]
Chemical Entities of Biological Interest	CHEBI	https://www.ebi.ac.uk/chebi/ (accessed on 28 April 2022)	[17]
NCBI organismal classification	NCBITaxon	https://github.com/obophenotype/ncbitaxon (accessed on 28 April 2022)	
Information Artifact Ontology	IAO	https://github.com/information-artifact-ontology/IAO/ (accessed on 28 April 2022)	
Units of measurement ontology	UO	https://github.com/bio-ontology-research-group/unit-ontology (accessed on 28 April 2022)	[18]
Ontology of Microbial Phenotypes	OMP	https://microbialphenotypes.org/wiki/index.php?title%20=%20Main_Page (accessed on 28 April 2022)	[19]
NCI Thesaurus OBO Edition	NCIT	https://github.com/NCI-Thesaurus/thesaurus-obo-edition (accessed on 28 April 2022)	
Uberon multi-species anatomy ontology	UBERON	http://obophenotype.github.io/uberon/ (accessed on 28 April 2022)	[20]
Phenotype And Trait Ontology	PATO	https://github.com/pato-ontology/pato/ (accessed on 28 April 2022)	
Human Phenotype Ontology	HP	https://hpo.jax.org/app/ (accessed on 28 April 2022)	[21]

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
