# Peer review of "Semantics of Dairy Fermented Foods: A Microbiologist’s Perspective"

_foods, 2022, doi:10.3390/foods11131939_

Round 1
Reviewer 1 Report
The article entitled “Semantics of dairy fermented foods: a microbiologist’s
perspective” is well written, but a few comments need to be addressed for acceptance.
Page 2: “We focused our analysis…………. (Parmigiano Reggiano)”- Reference missing. Why only these two fermented foods?
Page 4: Result- “starter culture for food fermentation”- Is any specific microbial starter culture used in this study?
Page 6: Result- “autochthonous food microbiota”- If this microbiota is already available, then why do we need any additional starter culture? Why can't we enhance these microbial communities for better yield in the fermentation process? Is there any specific reference for these?
GO pathways with CHEBI chemical entities-Reference missing?
Page 14: “(55 °C and 10 minutes, respectively).” - Any citation for this process?
What will be the impact of this temperature on the starter culture?
Author Response
C1: Page 2: “We focused our analysis…………. (Parmigiano Reggiano)”- Reference missing. Why only these two fermented foods?
R1: We thank the Reviewer for the comment: we have added one generic reference concerning dairy food consumption and diffusion, and two other more specific references for kefir and Parmigiano Reggiano. We have also rephrased the sentences, hoping that it is now clear that we chose these two dairy foods as representative of unripened and ripened fermented products.
C2: Page 4: Result- “starter culture for food fermentation”- Is any specific microbial starter culture used in this study?
R2: We thank the Reviewer for the comment: the study presented in this manuscript did not include any experimentation, but rather it is aimed at the organization of knowledge in the dairy fermentation domain. For this reason, the "starter culture for food fermentation” refers to the very concept of a starter culture, setting a specific definition for the term/concept, and connecting it to the general context of the other concepts in the dairy fermentation domain, and not to any specific use or instance of the term. We hope that we have clarified.
C3: Page 6: Result- “autochthonous food microbiota”- If this microbiota is already available, then why do we need any additional starter culture? Why can't we enhance these microbial communities for better yield in the fermentation process? Is there any specific reference for these?
R3: We thank the Reviewer for the comment: indeed, the "starter culture for food fermentation" (with definition “microbial culture responsible for the beginning of the fermentation process”) class has three sub-classes: " autochthonous food microbiota", "commercial starter culture", and "natural starter culture". In the model, we specified that the class "starter culture for food fermentation” participates in the "glucose metabolism during food fermentation"; as in the formalization of an ontology, sub-classes have the same annotation as super-classes, this means that formally each of the three different kinds of starter culture that we defined (autochthonous, commercial, and natural) can be used as starter culture, and that it depends on the specification of the fermented food production process. The comment of the Reviewer is pertinent as fermentation can effectively start from the autochthonous microbiota, and that is the case, for example, of spontaneously fermented foods or of fermented foods obtained by backslopping procedures. It is also true that often the autochthonous microorganisms are isolated from traditional fermented products and characterized in terms of favourable technological traits to be selected and used to improve the fermentation performance. There are a lot of references about this, however, to our opinion, they are out of context in the present manuscript. The model we included in the ONS ontology formally allows and recognizes that a fermentation process (specifically, the glucose metabolism during a fermentation process, which is among the initial steps of food fermentation) can start from an autochthonous food microbiota. We hope that we have addressed the questions raised.
C4: GO pathways with CHEBI chemical entities-Reference missing?
R4: We thank the Reviewer for the comment: all the references for external ontologies such as GO and CHEBI are reported in Table 1. By doing this, we aimed at just using prefixes for the various ontologies in the text, to keep it tidier, while giving credit to all the ontologies in a summary table clearly reporting the name of the ontology, its prefix and the reference. We hope that we have raised the question.
C5: Page 14: “(55 °C and 10 minutes, respectively).” - Any citation for this process? What will be the impact of this temperature on the starter culture?
R5: Thank you for the comment: we have added a reference. Concerning the question, yes, the cooking process does have an impact on the starter culture: we explained this aspect at the beginning of the paragraph. We hope that we have now clarified.
Reviewer 2 Report
The manuscript is interesting to read and shows extensive work plan. However, I suggest minor corrections:
Title illustrates ‘dairy fermented foods’ but it seems the work plan is focused in ‘fermented foods’ in general. Check it.
Check Table 1 last column for gene oncology. It needs clarification.
Add few more suitable references in Introduction and Discussion section.
In my opinion, future perspective of this work should be written more clearly with few more extra points in Conclusion section.
References should be formatted as per the journal instructions.
Author Response
C1: Title illustrates ‘dairy fermented foods’ but it seems the work plan is focused in ‘fermented foods’ in general. Check it.
R1: We thank the Reviewer for the comment: some of the concepts included in the ontology presented in this manuscript could be indeed considered generally connected to any fermentation process. Specifically, the concept regarding inoculum and starter culture can surely be generally applied to the fermentation domain and are not limited to the dairy fermented foods. Nevertheless, we extensively used two dairy fermented foods (Kefir and Parmigiano Reggiano) as use case for the modelling herein presented. For this reason, even though the future perspectives of this work would be more generally focused on food fermentation, we think that this manuscript is more precisely focused on dairy fermentation and prefer to keep the title as it is. We welcomed the suggestion of this Reviewer, and the suggestion in comment number 4, and expanded the future perspectives in the conclusion section.
C2: Check Table 1 last column for gene oncology. It needs clarification.
R2: We thank the Reviewer for the suggestion: we have added two references (Ashburner et al., 2000; Gene Ontology Consortium, 2021), according to the indications in the GeneOntology citation policy (http://geneontology.org/docs/go-citation-policy/)
C3: Add few more suitable references in Introduction and Discussion section.
R3: We thank the Reviewer for the suggestion: we have added additional references both in the Introduction and in the Discussion
C4: In my opinion, future perspective of this work should be written more clearly with few more extra points in Conclusion section.
R4: We thank the Reviewer for this suggestion: we added some more details on the future perspectives of this work, as also illustrated in the response to comment C1.
C5: References should be formatted as per the journal instructions.
R5: We apologize, we have now formatted the references according to journal’s style.
Reviewer 3 Report
Abstract section
1. The abstract need a lots of tremendous improvement and I therefore suggest that the abstract should be rewritten in the following pattern to illuminate on key finding from this study.
a. Brief introduction
b. Aim and objectives
c. Key and most important results
d. Conclusion and contribution to knowledge.
e. Please provide a more quantitative data rather more descriptive data.
Specifically,
1.Please kindly recast this sentence and replace the word ‘’allowing’’
‘’Food ontologies are acquiring a central role in human nutrition, allowing for a proper description of intervention and observational trials.’’
2.Please don’t start a sentence with ‘’while’’
‘’While most foods are consumed after several chemical, chemico-physical, or biological processes, leading to bioactive molecules, fermented foods provide the host with live microorganisms, thus carrying potential “genetic/functional” nutrients.’’
3 What do you mean by to date or till- date please kindly cross check
‘’To date, a proper ontology to structure and formalize the concepts used to describe fermented foods is lacking. ‘’
4. Please kindly recast the whole sentence
‘’Here we describe a semantic representation of concepts revolving around what consuming fermented foods entails, both from a technical and health point of view; actions were focused on Kefir and Parmigiano Reggiano.’’
5. What do you mean by ‘’In this way’’, please kindly replace
‘’In this way, we demonstrate the potential of ontologies to formalize the various gene pathways involved in raw ingredient transformation, connect them to resulting metabolites, and finally to their consequences on the fermented product, including technological, health and sensory aspects.’’
6. Please recast this sentence
‘’ Our work will be instrumental to the FAIR annotation of the experimental data in public repositories. This process will improve the findability, comparability and integration of heterogeneous data and the exploitation of publicly available datasets through meta-analyses.’’
Introduction
7. Please kindly recast this sentence and what do you mean by ‘’exert’’
‘’Foods exert benefits (or eventually damage) to human physiology that are greater than the benefits of the sums of their ingredients with the nutrients, elements, and other molecules therein included’’
8. Please kindly recast this sentence and what do you mean ‘’diffused food’’
‘’One of the most ancient and diffused food modification processes is represented by fermentation, in which the biological action of microorganisms is responsible for the food modifications.’’
9. Please provide a section that will provide a clear justification and problem statement for this study.
10. Please kindly provide a clear aim for this study because the one you provided is not clear enough
‘’Our aim was to provide an ontological resource that includes, in addition to conceptualization, the basis for the integration of data in nutritional studies, increasingly characterized by the use of multi-omics techniques’’.
Discussion section
1. The author need to relate the results obtained during this study with relevant discussion and compare the results obtained to previously results from other researchers
2. The manuscript is based on a very good concept methodologically executed but poorly written. The methods failed to align with the result with the discussion. Authors need to surrender this paper to serious editorial review by an English expert or language skilled colleague. This would illuminate the manuscript and makes it more comprehensive.
References
The authors should check the reference if they are in accordance with the format stipulated by the journal.

Author Response
C1: Abstract section
The abstract need a lots of tremendous improvement and I therefore suggest that the abstract should be rewritten in the following pattern to illuminate on key finding from this study.
- Brief introduction
- Aim and objectives
- Key and most important results
- Conclusion and contribution to knowledge.
- Please provide a more quantitative data rather more descriptive data.
R1: We thank the Reviewer for the comments and suggestions: we tried to do our best to address all the points by rephrasing the sentences, also according to the specific points below (C2-C7). According to the Journal’s Guideline, the maximum length of the abstract is 200 words, which means that it must be very concise. We hope that now the abstract is improved.
We would also like to point out that the present manuscript did not include any experimental work, rather it is aimed at the organization of knowledge in the dairy fermentation domain. For this reason, we cannot provide quantitative data to fulfil point e.) of the suggestions, as there are none.
C2: Please kindly recast this sentence and replace the word ‘’allowing’’ ‘’Food ontologies are acquiring a central role in human nutrition, allowing for a proper description of intervention and observational trials.’’
R2: We thank the Reviewer for the suggestion: we have rephrased the sentence and we hope that is now clear.
C3: Please don’t start a sentence with ‘’while’’ ‘’While most foods are consumed after several chemical, chemico-physical, or biological processes, leading to bioactive molecules, fermented foods provide the host with live microorganisms, thus carrying potential “genetic/functional” nutrients.’’
R3: We thank the Reviewer for the suggestion: we have rephrased the sentence.
C4: What do you mean by to date or till- date please kindly cross check ‘’To date, a proper ontology to structure and formalize the concepts used to describe fermented foods is lacking. ‘’
R4: With “to date” we mean “up until the present time”. We hope we have clarified the point.
C5: Please kindly recast the whole sentence ‘’Here we describe a semantic representation of concepts revolving around what consuming fermented foods entails, both from a technical and health point of view; actions were focused on Kefir and Parmigiano Reggiano.’’
R5: We thank the Reviewer for the suggestion: we have modified the sentence and we hope to have improved the content.
C6: What do you mean by ‘’In this way’’, please kindly replace ‘’In this way, we demonstrate the potential of ontologies to formalize the various gene pathways involved in raw ingredient transformation, connect them to resulting metabolites, and finally to their consequences on the fermented product, including technological, health and sensory aspects.’’
R6: Thank you for the useful suggestions: we have modified the text and we hope that now it is improved.
C7: Please recast this sentence ‘’ Our work will be instrumental to the FAIR annotation of the experimental data in public repositories. This process will improve the findability, comparability and integration of heterogeneous data and the exploitation of publicly available datasets through meta-analyses.’’
R7: Thank you for the useful suggestions: we improved the sentence. Reference to the FAIR data principles were removed, to avoid using an acronym in the abstract section.
C8: Please kindly recast this sentence and what do you mean by ‘’exert’’ ‘’Foods exert benefits (or eventually damage) to human physiology that are greater than the benefits of the sums of their ingredients with the nutrients, elements, and other molecules therein included’’
R8: We thank the Reviewer for the suggestion: with “exert” we meant “to put forth/to put into action”. We meant that consumed food has effects on human physiology and health which can’t be simply explained by considering the effect that each ingredient would have, as during food preparation a series of transformation occur. We modified the sentence changing “exert” with “yield” for a better understanding.
C9: Please kindly recast this sentence and what do you mean ‘’diffused food’’ ‘’One of the most ancient and diffused food modification processes is represented by fermentation, in which the biological action of microorganisms is responsible for the food modifications.’’
R9: We thank the Reviewer for the comment. We have changed “diffused” with “widespread”.
C10: Please provide a section that will provide a clear justification and problem statement for this study.
We thank the Reviewer for the suggestion: please, see our answer to point C11.
C11:. Please kindly provide a clear aim for this study because the one you provided is not clear enough ‘’Our aim was to provide an ontological resource that includes, in addition to conceptualization, the basis for the integration of data in nutritional studies, increasingly characterized by the use of multi-omics techniques’’.
R10 - R11: We thank the Reviewer for the comment. In the introduction, we also stated our aim few lines above the paragraph highlighted by the Reviewer. We acknowledge that a formal problem statement (C10) and a clear objective (C11) was missing, and rephrased the paragraph.
C12: The author need to relate the results obtained during this study with relevant discussion and compare the results obtained to previously results from other researchers
R12: We thank the Reviewer for the comment. As indicated in answer to comment C1, the present manuscript did not include any experimentation, and is rather aimed at the organization of knowledge in the dairy fermentation domain. For this reason, our results are represented by the very classes and connection defined and modelled, as included in the latest release version of ONS. We indeed relate our work to previous work by other researchers, not in a comparison, but rather by acquiring the same solutions. As described in the methods and reported briefly in Table 1, in fact, it is common and desirable practice to include avoid re-defining classes/concept that are already defined in other public ontologies. This concept is often referred to with “avoid re-inventing the wheel” in ontologies development and illustrates how important it is to start from the work of others as a basis, rather than re-defining common concepts. We hope that we have provide a convincing explanation.
C13: The manuscript is based on a very good concept methodologically executed but poorly written. The methods failed to align with the result with the discussion. Authors need to surrender this paper to serious editorial review by an English expert or language skilled colleague. This would illuminate the manuscript and makes it more comprehensive.
R13: We thank the Reviewer for the comment: the whole manuscript was revised for improvement of written English.
C14: References The authors should check the reference if they are in accordance with the format stipulated by the journal.
R14: We apologize, we have now formatted the references according to journal’s style.